# Vaginal and Cervical Microbiota Composition in Patients with Endometrial Cancer

**DOI:** 10.3390/ijms24098266

**Published:** 2023-05-05

**Authors:** Bartłomiej Barczyński, Karolina Frąszczak, Ewelina Grywalska, Jan Kotarski, Izabela Korona-Głowniak

**Affiliations:** 11st Department of Oncological Gynaecology and Gynaecology, Medical University in Lublin, 20-081 Lublin, Poland; bbarczynski@poczta.onet.pl; 2Department of Experimental Immunology, Medical University in Lublin, 20-093 Lublin, Poland; ewelina.grywalska@umlub.pl; 3Independent Laboratory of Cancer Diagnostics and Immunology, Medical University in Lublin, 20-093 Lublin, Poland; jan.kotarski.gabinet@gmail.com; 4Department of Pharmaceutical Microbiology, Medical University in Lublin, 20-093 Lublin, Poland

**Keywords:** microbiome, endometrial cancer, *Lactobacillus*, *Mobiluncus*, *Dialister*

## Abstract

According to recent data, changes in the vaginal microbiota could affect the risk of gynaecological cancers. Women suffering from endometrial cancer present significant changes in cervicovaginal microbiota composition. The objective of our study was to characterize the cervicovaginal microbiota of women undergoing hysterectomy due to benign disease, atypical hyperplasia, and endometrial cancer; The study included 96 patients, who undergone surgical treatment due to benign uterine disease, precancerous endometrial lesion, and endometrial cancer. Quantitative and qualitative real-time PCR analysis of DNA isolated from vaginal fornix and endocervical canal samples was performed to detect the 19 most commonly identified microorganisms, including different *Lactobacillus* spp., *Atopobium*, *Bifidobacterium*, *Chlamydia*, and *Gardnerella*; At least one of the tested microorganisms was identified in 88.5% of vaginal and 83.3% of cervical samples. *Lactobacillus iners* was significantly more frequent in patients with benign condition, whereas *Dialister pneumosintes* and *Mobiluncus curtisii* was more frequent in cancer patients; *Mobiluncus curtisi* and *Dialister pneumosintes,* which were identified as significantly more common in endometrial cancer vaginal samples, may be considered as potential endometrial cancer co-factors which promote/stimulate carcinogenesis. However, the exact mechanism of such activity remains unexplained and requires further investigations.

## 1. Introduction

The female genital tract is colonized by a diverse community of various commensal, symbiotic and pathogenic microorganisms, which include bacteria, archaea, protozoa, fungi, and viruses. All these microorganisms form a specific microbiome and play an essential role in maintaining women’s health and homeostasis. It has to be noted that according to some microbiome researchers, the term “microbiome” should be restricted to studies on bacterial communities, and studies characterizing viruses and fungi should use terms “virome” and “mycobiome”, respectively [1]. In the majority of healthy women the microbiome of the lower part of female genital tract (vagina and uterine cervix) tends to be dominated by different *Lactobacillus* species and diverse additional anaerobic taxa [2]. The upper part of the female genital tract (uterine corpus, fallopian tubes, ovaries) is considered as sterile and/or should contain considerably lower number of microbiota in healthy women [1]. The lactobacilli colonizing vagina and uterine cervix are believed to benefit the host by lowering vaginal pH through fermentation end products, thereby reducing the likelihood of allochthonous microbial colonization or pathogen invasion [2].

Recent studies have proved that changes in the vaginal microbiota could affect the risk of gynaecological cancers, as well as other gynaecological non-malignant conditions [3]. Women suffering from endometriosis [4], chronic endometritis [5], endometrial polyps [6], dysfunctional uterine bleeding [7], endometrial cancer/hyperplasia [8], or infertility [9] have a changed composition of the vaginal microbiota in comparison with healthy individuals. The vaginal ecosystem is a complicated microbial niche which allows for survival and proliferation of a number of both beneficial bacteria as well as opportunistic pathogens. Any pelvic inflammatory process caused by polymicrobial infection resulting in inflammation of endometrium and/or adnexa, may be considered a carcinogenic factor, stimulating/intensifying carcinogenesis by microbial dysregulation, which is also a conceivable hypothesis [10]. 

Even though a disruption of the vaginal microbiota may potentially promote gynaecologic carcinogenesis, the specific role of the microbiota in gynaecologic malignancies remains unclear [3]. Furthermore, endometrial cancer is promoted by obesity, hormonal imbalances, diabetes and metabolic syndrome, all of which may promote changes in the microbiota [11,12,13]. 

The objective of this study was to characterize the cervicovaginal microbiota of women undergoing hysterectomy due to benign disease, atypical hyperplasia, and endometrial cancer. We hypothesized that there is a microbiota factor that may be a marker in patients diagnosed with the cancer disease.

## 2. Results

### 2.1. Study Population

A total of 96 patients with an age range of 44–86 years (mean 60.1 ± 9.8) were recruited in our research. In selected patients, cancer (48 patients), hyperplasia (21 patients), and benign condition (27 patients) were diagnosed. The demographic and clinical characteristics for all participants are listed in Table 1. Patients diagnosed with endometrial cancer were significantly older, predominantly postmenopausal.

### 2.2. Qualitative Identification of Vaginal/Cervical Microbiota

In total, 192 paired vaginal and cervical samples were analysed. In molecular analysis with real-time PCR, from the 96 patients studied, 557 species/genes of 19 various microbial species (*Atopobium vaginae*, *Dialister pneumosintes*, *Bifidobacterium bifidum*, *Bifidobacterium breve*, *Bifidobacterium longum*, *Candida glabrata*, *Candida albicans*, *Chlamydia trachomatis*, *Gardnerella vaginalis*, *Lactobacillus acidophilus*, *Lactobacillus crispatus*, *Lactobacillus gasseri*, *Lactobacillus iners*, *Lactobacillus jensenii*, *Lactobacillus vaginalis*, *Sneathia sanguinegens*, *Streptococcus agalactiae*, *Mobiluncus curtisii and Fusobacterium nucleatum*) were retrieved: 322 from vaginal and 235 from cervical samples. In one sample, 0–7 species/genes were detected. Eighty-five (88.5%) vaginal and eighty (83.3%) cervical samples were positive for at least one of the tested microorganisms.

### 2.3. Quantitative Identification of Vaginal/Cervical Microbiota

Difference in mean numbers of bacterial species detected in one sample in vaginal (3.35 ± 1.9, range 0–7) and cervical (2.44 ± 1.9, range 0–7) samples was observed with statistical significance (*p* < 0.0001). The statistical significance of mean bacterial frequencies between vaginal and cervical samples were detected in each tested group of patients (Table 2). Moreover, there were also significant differences between bacterial frequencies in vaginal samples of cancer samples vs. hyperplasia samples and cancer samples vs. benign samples (*p* = 0.0005 and *p* = 0.003, respectively) as well as between cervical samples in those groups (*p* = 0.0006 and *p* = 0.0005, respectively).

The most frequent bacterial species in all groups were *G. vaginalis*, *F. nucleatum*, *M. curtisi*, *L. gasseri*, *L. iners*, *L. crispatus* (Figure 1). Amongst species tested, the frequency of *F. nucleatum* and *M. curtisii* in vaginal samples were significantly higher in comparison with cervical samples (38.5% vs. 24.0%, *p* = 0.043 and 35.5% vs. 17.7%, *p* = 0.009, respectively). None of the samples were positive for *C. trachomatis* and *L. acidophilus*.

### 2.4. Vaginal/Cervical Microbiota Composition in Patients with Endometrial Cancer and Precancerous Lesion

There were no associations between the frequency of tested bacteria and the malignancy disease. However, the statistical analysis revealed the differences in prevalence of several species in patients with different diagnoses. Figure 2 presents bacterial frequency analysis in the cancer, hyperplasia and benign groups.

*L. iners* was significantly more frequent in patients with benign conditions, whereas *D. pneumosintes* and *M. curtisii* was more frequent in cancer patients. In general, *Lactobacillus* species and *G. vaginalis* were detected significantly less frequently in the cancer group of patients.

### 2.5. Vaginal/Cervical Microbiota Composition in Regard to Menopausal Status

In postmenopausal women, significantly higher prevalence of *M. curtisii* in vaginal sample was observed (Table 3). However, the greater prevalence of *F. nucleatum* in this group was not significant. *Lactobacillus* spp. was less frequent in the postmenopausal group with statistical significance for *L. gasseri, L. jensenii* and *L. iners* as well as *B. bifidum*. Moreover, significance was found for the higher prevalence of *Atopobium vaginae* and *G. vaginalis* in group of pre- and perimenopausal women.

### 2.6. Relative Profiling of Vaginal/Cervical Microbiota Composition in Endometrial Cancer vs. Atypical Hyperplasia or Benign Uterine Condition

Microbial profiles showing bacterial composition and relative abundance of vaginal and cervical samples are presented in Figure 3. The ΔΔCT method was used for the relative profiling and comparison between two populations from the cancer group and benign or hyperplasia group (Figure 3). Microbial DNA qPCR Array correlated increased amounts of *L. iners* and *G. vaginalis* in vaginal samples of patients with cancer, with reduced amounts of *M. curtisii*, *F. nucleatum* (Figure 3). The relative abundance of Pan Bacteria significantly decreased in cancer group patients both in vaginal and cervical samples.

### 2.7. Prediction of Bacterial Communities Profile

Principal component analysis (PCA) was applied to compare the overall structure of vaginal and cervical microbiota of all samples using data scaled to UV (Figure 4). The built model explains 43.0% of the variations. The first principal component explained 22.7% of the overall variability, whereas the second and third principal components explained 11.0% and 10% of variability. As shown in Figure 4, most observations are close to the plot origin showing rather average properties. The first PC was significantly positively correlated with *G. vaginalis*, *C. glabrata*, *C. albicans*, *L. vaginalis*, *L. jensenii* and *B. longum*. The significant positive correlation of the second PCs in with *L. iners*, *L. crispatus* and *F. nucleatum* was observed. PCA identified four distinct clusters of microbiota profiles correlating strongly with a predominance of species, respectively, as well as connected by a group of community profiles representing mixed microbiota. 

Clustering by disease status was not observed for the tested vaginal/cervical samples. Hierarchical cluster analysis based on the relative abundance of species revealed a separation of four groups of samples (red, blue, green and yellow groups, Figure 4) on the basis of the first two principal component (PC) scores. This discrimination was also confirmed by discriminant analysis (PLS-DA). According to PLS-DA, species such as *B. bifidum*, *B. longum*, *C. glabrata* (vaginal samples), *C. albicans*, *L. gasseri*, *L. jensenii,* and *L. vaginalis* were markedly high in the green group. This group consisted of women with hyperplasia and benign lesions mainly. The blue group with cancer patients mainly has an abundance of *A. vaginae, C. glabrata* (cervical samples), *C. albicans*, *G. vaginalis*, *S. sanguinegens*, *F. nucleatum* along with *Lactobacillus* spp. negatively correlated. The red group containing a majority of cancer patients was markedly colonized by low amount of bacteria tested reported by generally negative association of *G. vaginalis, L. crispatus*, *L. gasseri*, *L. iners*, *A. vaginae* and *B. breve*. *L. crispatus, L. gasseri, L. iners, S. agalactiae* and *B. breve* contributed mainly in the yellow group of patients *but L. jensenii, L. vaginalis* and *M. curtisii* were negatively associated. The yellow group comprised of non-cancer patients mainly.

## 3. Discussion

In the present study, a pilot high-throughput microbiota assessment of the lower part of the female reproductive tract in patients diagnosed with endometrial cancer, atypical hyperplasia and benign uterine conditions was performed. As far as we know, this is the first original report on the abundance of specific species of bacteria, most commonly identified in cervicovaginal swabs.

The vaginal microbiota includes a diverse range of bacterial species, usually between 20 and 140 in any particular individual, with observed variation between individuals and over time [12,13]. Thus, in recent years, it has been hypothesized that there is no “single core” microbiome for human vagina, and it seems to change throughout women’s lifetime and depends on menstrual status, individuals’ habits and changes in diet. It has also been speculated that the vaginal microbiome may be governed by genetically determined factors, as there are significant changes in bacterial communities between healthy women from different ethnic groups [14]. The vaginal microbiome in healthy asymptomatic women can be classified according to the most dominant species of bacteria present, which is most commonly a species of *Lactobacillus* [15,16]. *L. crispatus*, *L. iners*, *L. gasseri*, and *L. jensenii* were dominant lactic acid bacteria identified in asymptomatic North American women, whereas the last group of bacteria community was dominated by higher proportions of strictly anaerobic organisms, including *Prevotella*, *Dialister*, *Atopobium*, *Gardnerella*, *Megasphaera*, *Peptoniphilus*, *Sneathia*, *Eggerthella*, *Aerococcus*, *Finegoldia*, and *Mobiluncus* spp. [16]. In our present study, the vaginal and cervical microbiome of healthy women consisted mainly of *L. crispatus*, *L. iners*, *L. gasseri*, and *G. vaginalis*, whereas *M. curtisii* and *F. nucleatum* were mostly identified in women with endometrial cancer. Of note, cervical and vaginal microbiome was actually correlated, suggesting that vaginal sampling is equally as indicatory as endocervical sampling.

The first report on possible role of overall female genital tract microbiome on manifestation or etiology of endometrial cancer was presented in 2016 by Marina Walther-António et al., who demonstrated that some specific bacteria species from Firmicutes, Spirochaetes, Actinobacteria, Bacteroidetes, and Proteobacteria taxa were identified significantly more common in patients with endometrial cancer [17]. The recognition of *A. vaginae* and *Porphyromonas* spp. in samples from the lower genital tract was strongly associated with the presence of endometrial cancer [17]. Gressel at al., who characterized the microbiota of postmenopausal women undergoing hysterectomy for endometrial cancer, also proving the microbial diversity of different anatomic niches of endometrial cancer patients compared with non-malignant controls [18]. Furthermore, the composition of the cervicovaginal microbiota varied among different pathological endometrial cancer subtypes [18]. In a study of Hakimjavadi et al., *F. ulcerans* and *Prevotella bivia* species were significantly more abundant in women with high-grade endometrial carcinoma (grade 3 endometrioid, serous, small cell, clear cell, undifferentiated, or dedifferentiated carcinoma, uterine carcinosarcoma) compared with low-grade endometrial cancers [19]. Moreover, *Fusobacterium ulcerans* was the only species significantly more abundant in high-grade endometrial cancer patients compared with women with benign endometrial histology [19]. On the other hand, low-grade endometrial cancer patients showed distinct microbiome abundance of *Clostridium* spp., *Corynebacterium amycolatum*, *Lactobacillus gasseri*, and *Peptoniphilus duerdeni*, when compared to women with benign diseases [19]. Similarly, there were significant changes in relative Bacterioidetes and Lactobacilli abundance in regard to cancer clinical stage, i.e., notably increased Bacterioidetes abundance in FIGO stage IB/II vs. stage IA and low lactobacilli in all stages of the disease [20].

The mechanisms by which the microbiome may influence endometrial cancer pathogenesis still have to be explicated but seem to be multifactorial in the context of alterations in cancer cell signaling pathways. It is commonly known that the majority of endometrial cancers show estrogen-dependent proliferation, but the carcinogenic mechanisms are not completely explained except for some specific single oncogenes and tumor suppressor genes (e.g., p53) mutations. Environmental and host factors such as obesity, diabetes mellitus and certain estrogen/progesterone hormonal changes do not thoroughly explain tumorigenic mechanism. Overall microbiome imbalance may serve as an important risk factor for cancers. It has been proved that the pathomechanism of gastric cancer is associated with *Helicobacter pylori* infection [21]. Similarly, *F. nucleatum* bowel colonization may serve as a key pathogenic factor for colorectal cancer [22]. Thus, we hypothesize that alterations in vaginal microbiome may also play an important role in endometrial cancer tumorigenesis. In our analyses, we confirmed significant cervicovaginal dysbiosis in endometrial cancer patients. Moreover, *M. curtisi* and *D. pneumosintes,* which were identified as significantly more common in endometrial cancer patients, may be considered as potential endometrial cancer co-factors. However, this hypothesis requires further investigations. The most important question is, what is the possible mechanism of such co-factoring mechanism in endometrial tumorigenesis?

Unopposed estrogen access is one of the key risk factors of endometrial hyperplasia and cancer [23]. The human gut microbiome impacts estrogen levels through the secretion of β-glucuronidases and β-glucuronides, enzymes involved in estrogen de-conjugation and conjugation [24]. Thus, it influences endogenous estrogen metabolism by modulating the enterohepatic circulation of estrogens, affecting circulating and excreted estrogen levels [24]. Promoting estrogen metabolite deconjugation reactions may result in increased reabsorption of free estrogens and, as a consequence, increase women’s lifetime burden of estrogen exposure [24]. Such β-glucuronidase activity was confirmed in many bacterial species from Firmicutes and Bacterioidetes phyla, mostly colonizing lower female genital tract too [25,26,27]. It has been proved that altered Firmicutes/Bacteroidetes ratio leads to increased deconjugated estrogen levels and increased circulating free estrogen increases binding to estrogen receptors (α and β), causing development/progression of estrogen-dependent diseases, e.g., endometrial and breast cancers or endometriosis [28].

Epidemiological evidence confirms that carcinogenesis in endometrial cancer may be promoted by an inflammatory milieu [29]. Chronic inflammation might mediate the association between obesity and endometrial cancer [29]. A significant increase in risk of endometrial cancer was observed with elevated levels of C-reactive protein, interleukin 6 (IL-6), and interleukin 1 receptor antagonist (IL-1Ra) [29]. Thus, much effort is devoted to proving a relationship between microbiome dysbiosis and the inflammation. Very recently, a hypothesis was presented that some microbiota may stimulate the initiation of inflammation, induce immunopathological changes and subsequently stimulate carcinogenesis [30]. Such pro-carcinogenic properties linked to inflammatory responses have been described in relation to gut microbiota and colorectal cancer [30]. However, in a study by Lu et al., mRNA expression of pro-inflammatory cytokines (e.g., IL-6, IL-8, and IL-17) was found to be significantly increased in endometrial cancer women when compared to benign uterine lesion cohort [31]. According to Morselli et al., *G. vaginalis* strains isolated from women with bacterial vaginosis show potential to significantly upregulate the production of pro-inflammatory cytokines (especially IL-6 and IL-8) [32]. It is hypothesized that *G. vaginalis*-mediated epithelial immune response including nuclear factor kappa B (NF-κB) activation and pro-inflammatory cytokine release may be initiated partially through TLR2-dependent signaling pathways [33]. It has also been proved that *G. vaginalis* strains possess slight immune-stimulating activities against monocyte-derived dendritic cells and T cells, reflecting a defective inflammatory response and giving rise to an inflammatory clinical disease profile [34]. Interestingly, treatment with *Lactobacillus* spp. strains or their cell-free supernatants in response to *G. vaginalis* infection in a HeLa cell infection model resulted in decreased secretion of pro-inflammatory cytokines and decreased activation of NF-κB [35]. Similar pro-inflammatory potential was also observed in case of *A. vaginae* and *Sneathia amnii*, which exhibited the pro-inflammatory potential through induction of specific cytokines (e.g., IL-6, IL-8, interferon gamma-induced protein-10, monocyte chemotactic protein 10), inducible nitric oxide synthases (iNOS), and oxidative stress-associated compounds [36]. Such an inflammatory effect potentially favoring the onset and/or progression of endometrial cancer seems to be independent of ethnicity and was also observed in sub-Saharan African women [37].

Specific qualitative changes in local microbiome may also be linked to certain epigenetic modifications in gene functions through the process of DNA methylation, histone modification or non-coding RNA expression changes [38]. General epigenetic dysregulations have also been reported in regard to development and progression of endometrial cancer [39]. Witkin et al. proved that histone deacetylase-1 level in vaginal epithelial cells varied concerning microbiome *Lactobacilli* prevalence [40]. On the other hand, Anton et al. confirmed that *G. vaginalis* in an in vitro setting increased inflammatory-associated miRNA expression, which was reversed by *L. iners* [41]. However, neither of the studies cited above reports any data on possible link between local vaginal/cervicovaginal microbiome-induced epigenetic modifications and endometrial cancer development, and there are still no such data in the literature.

Recent observations suggest a strong relationship between vaginal microbiome composition and survival in cancer patients. It is commonly known that high-oncogenic risk HPV persistent infection contributes to the development of cervical cancer. However, it has been suggested that microbiome diversity may correspond to different survival of endometrial cancer patients [42]. One of the potential causes of higher mortality of Afro-American suffering from early-stage endometrial cancer may be the result of an increased abundance of Firmicutes and Cyanobacteria in the endometrial tumor microenvironment [42]. It has also been confirmed that the vaginal microbiome of women diagnosed with ovarian cancer contains a significantly lower number of lactobacilli strains [43]. Over one-third of ovarian cancer patients with primary platinum-resistant disease are presented with a vaginal microbiome dominated by *Escherichia* (>20% relative abundance), whereas only one out of 23 patients with platinum super-sensitive disease (platinum-free interval above 24 months) showed *Escherichia*-dominated microbiome [43]. Moreover, vaginal *L. iners* abundance was associated with little (<1 cm), or no, gross residual disease after primary debulking cytoreductive cancer surgery [43]. Thus, normal vaginal microbiome maintenance seems to be an important prophylactic goal for healthcare clinicians treating women with risk factors of any gynecologic malignancy. It has to be underlined that antibiotic use in women with vaginal microbiome disruptions undergoing oncologic treatment may not be effective, as Chamber et al. hypothesized that ovarian tumor-bearing antibiotic-treated mice exhibited accelerated tumor growth and resistance to cisplatin therapy due to reduced apoptosis, increased DNA damage repair, and enhanced angiogenesis [44].

## 4. Materials and Methods

### 4.1. Participants

This prospective study was conducted between 2018 and 2019 at the Ist Department of Oncological Gynaecology and Gynaecology, Medical University in Lublin, Poland. Patients meeting inclusion criteria were approached initially by a gynecologic oncologist. 

The inclusion criteria were: patients qualified for total/subtotal hysterectomy due to endometrial cancer, atypical hyperplasia, and benign uterine diseases (fibroids, CIN, dysfunctional uterine bleeding, ovarian cyst).

Exclusion criteria were: intravaginal infections within the last 3 months, allergy, autoimmune diseases, pregnancy, previous history of cancer disease.

The protocol was reviewed and approved by the Bioethics Committee of the Medical University in Lublin (KE-0254/79/2019) and performed in compliance with the Helsinki declaration. All patients gave their written informed consent to using their material for scientific purposes and signed the written acquiescence form to participate in the study.

### 4.2. Sample Collection

All the vaginal and cervical swabs were collected by the surgeon (with guidance on-site by the research team) immediately after the administration of anaesthesia and immediately preceding the standard pre-surgical betadine douche. Both the vaginal and cervical swabs were performed with two sterile Dacron swabs each and placed in a sterile tube with 1 mL of SLB buffer kept on dry ice until storage at −80 °C.

### 4.3. Real-Time PCR Analysis

The vaginal and cervical samples were stored at −80 °C until RT-PCR could be performed. DNA from samples were extracted using Genomic DNA purification with QIAamp DNA Mini Kit (Qiagen, Germantown, MD, USA) according to the manufacturer’s instructions and analysed with the Custom Microbial DNA qPCR Array (Qiagen, Germantown, MD, USA). Real-time PCR assays were performed (Light Cycler 96, Roche, Basel, Switzerland) using the 16S rRNA gene as the target and using PCR amplification primers and hydrolysis-probe detection, which increases the specificity of each assay. Each Microbial DNA qPCR Array plate analysed one sample for 19 species (NCBI Tax ID)/Gene at a time. Pan-Bacteria assays that detect a broad range of bacterial species were included to serve as positive controls for the presence of bacterial DNA (positive tests for species within these 7 phyla: Actinobacteria, Bacteroidetes, Euryarchaeota, Firmicutes, Fusobacteria, Proteobacteria, and Tenericutes). Relative profiling applications was measured for host genomic DNA and overall bacterial load. Inclusion of these analyses allows the user to normalize sample input using ΔΔCT.

### 4.4. Statistical Analysis

The statistical analysis was performed with Tibco Statistica 13.3 (StatSoft, Palo Alto, CA, USA). The values of the parameters were presented as medians, minimum, and maximum value. Normal distribution of continuous variables was tested using Shapiro–Wilk test. The Mann–Whitney U-test was used for independent variables comparisons. Kruskal–Wallis ANOVA and multiple comparisons of mean ranks (as post-hoc analysis) were applied for the analysis of differences between more than two groups. The power and direction of association between pairs of continuous variables (studied groups) were determined using Spearman’s coefficient of rank correlation. The distribution of discrete variables in groups was compared with the Pearson’s Chi-square test or the Fisher’s exact test. The multivariate data analyses were conducted using the SIMCA 16 (v16.0.2, Umetrics, Sweden). Relative bacterial species abundance in vaginal and cervical samples were calculated according to the real-time PCR data analyzing protocol [20]. Principal component analysis (PCA) was used for identifying similarities and differences between analyzed samples. Dataas were scaled to unit variance and centered. Hierarchical Cluster Analysis (HCA) and partial last square discriminant analysis model (PLS-DA) were used for vaginal sample classification and predictions.

## 5. Conclusions

Some bacterial strains, including *M. curtisi* and *D. pneumosintes*, which were identified as significantly more common in our study group of endometrial cancer vaginal samples, may be considered as potential co-factors which may play an important role in promoting/stimulating carcinogenesis. One of the possible explanations for this phenomenon is based on the hypothesis that some microbiota may influence inflammatory responses by upregulating the production of pro-inflammatory cytokines (especially IL-6 and IL-8) or impair inflammatory response by specific dendritic cells subsets immunomodulation.

## Figures and Tables

**Figure 1 ijms-24-08266-f001:**
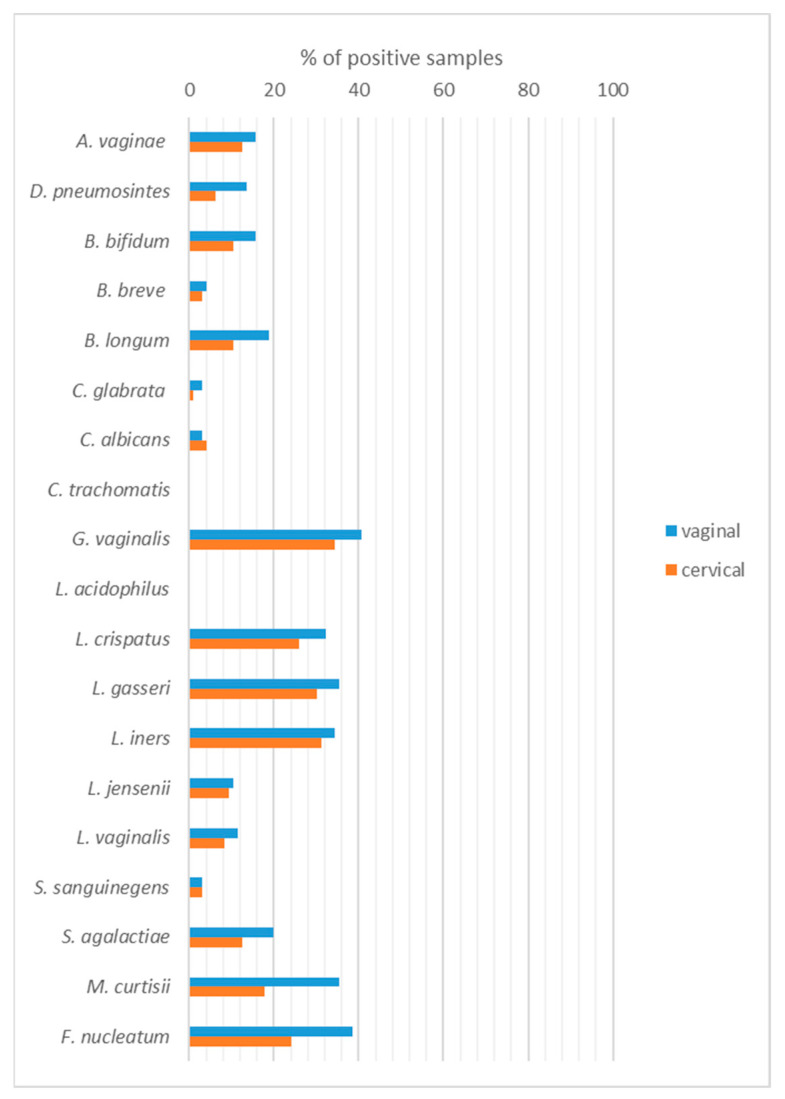
The distribution of microorganisms in the vaginal (V) and cervical (C) samples detected by molecular methods.

**Figure 2 ijms-24-08266-f002:**
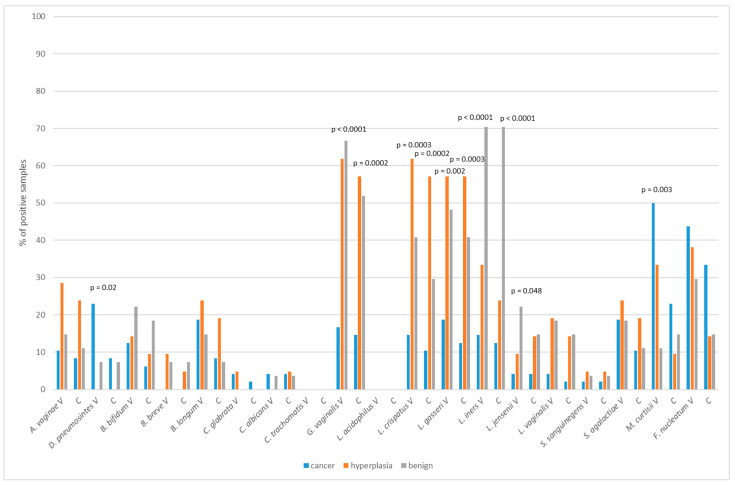
The distribution of microorganisms in the vaginal (V) and cervical (C) samples from patients with different entities obtained by molecular methods.

**Figure 3 ijms-24-08266-f003:**
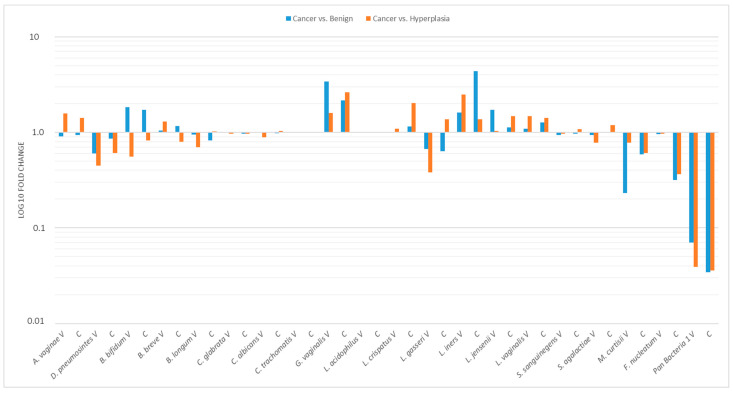
Accurate profiling of pathogenic and commensal microbes in patients with endometrial cancer, atypical hyperplasia, and benign uterine disease. Fold change in microbial species abundance in patients with different diagnoses (cancer and benign condition as well as cancer and hyperplasia groups) was calculated by the ∆∆CT method using human genomic DNA to normalize (HBB1 gene). At least a 5 to 10-fold increase or decrease in relative abundance may be considered significant. (V): vaginal; (C): cervical samples.

**Figure 4 ijms-24-08266-f004:**
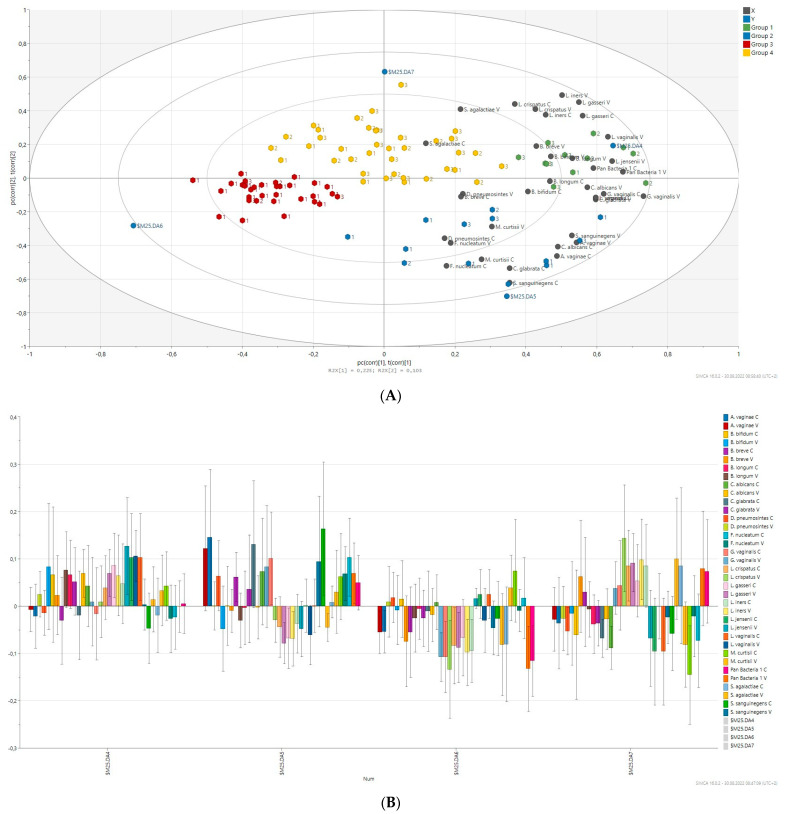
PLS-DA with vaginal (V)/cervical (C) samples was performed based on the 17 bacterial species tested. (**A**) Biplot of PLS-DA with vaginal samples. Different colours of points in plots were represented for different groups. (**B**) Coefficient overview plot displays how bacterial species take part in four created groups: M25DA4, green; M25DA5, blue; M25DA6, red; M25DA7, yellow.

**Table 1 ijms-24-08266-t001:** Demographic and clinical characteristics of the 96 women studied.

Variables	Total (*n* = 96)	Benign (*n* = 27)	Hyperplasia (*n* = 21)	Cancer (*n* = 48)	*p*-Value
Age (years) mean ± SD	60.1 ± 9.8	54.4 ± 9.0	55.6 ± 8.2	65.3 ± 8.2	<0.0001
BMI (mean ± SD)	30.5 ± 6.1	29.7 ± 5.1	29.3 ± 5.8	31.5 ± 6.6	0.31
Menopausal status					<0.0001
Pre	26 (27.1)	15 (55.6)	8 (38.1)	3 (6.3)	
Peri	9 (9.4)	3 (11.1)	5 (23.8)	1 (2.1)	
Post	61 (63.5)	9 (33.3)	8 (38.1)	44 (91.7)	
Histotype		NA	NA		
Endometroid	40 (41.7)			40 (83.3)	
Serous	3 (3.1)			3 (6.3)	
Carcinosarcoma	4 (4.2)			4 (8.3)	
Clearcell	1 (1.0)			1 (2.8)	
Grade		NA	NA		
G1	12 (12.5)			12 (25.0)	
G2	28 (29.2)			28 (58.3)	
G3	6 (6.3)			6 (12.5)	
unknown	2 (2.1)			2 (4.2)	
FIGO		NA	NA		
1A	25 (26.0)			25 (52.8)	
1B	16 (16.7)			16 (33.3)	
2	4 (4.2)			4 (8.3)	
3A	1 (1.0)			1 (2.1)	
3B	2 (2.1)			2 (4.2)	

NA—not applicable.

**Table 2 ijms-24-08266-t002:** Mean numbers of bacterial species in one sample in vaginal and cervical samples from different entities.

Entity	Vaginal Samples	Cervical Samples	*p*-Value
Cancer	2.6 ± 1.9	1.6 ± 1.6	<0.0001
Hyperplasia	4.2 ± 1.2	3.3 ± 1.8	0.034
Benign	4.0 ± 1.9	3.2 ± 1.8	0.0055

**Table 3 ijms-24-08266-t003:** The distribution of microorganisms in the vaginal (V) and cervical (C) samples from patients with different menopausal state obtained by molecular methods.

Microorganisms		Pre- and Perimenopausal (*n* = 35)	Postmenopausal (*n* = 61)	*p*-Value
*A. vaginae*	V	11 (31.4%)	4 (6.6%)	0.002
	C	8 (22.9%)	4 (6.6%)	0.024
*D. pneumosintes*	V	3 (8.6%)	10 (16.4%)	0.22
	C	3 (8.6%)	3 (4.9%)	0.38
*B. bifidum*	V	9 (25.7%)	6 (9.8%)	0.040
	C	7 (20.0%)	3 (4.9%)	0.026
*B. breve*	V	1 (2.9%)	3 (4.9%)	0.54
	C	1 (2.9%)	2 (3.3%)	0.70
*B. longum*	V	7 (20.0%)	11 (18.0%)	0.51
	C	5 (14.3%)	5 (8.2%)	0.27
*C. glabrata*	V	1 (2.9%)	2 (3.3%)	0.70
	C	0 (0)	1 (1.6%)	0.64
*C. albicans*	V	2 (5.7%)	1 (1.6%)	0.30
	C	2 (5.7%)	2 (3.3%)	0.46
*G. vaginalis*	V	25 (71.4%)	14 (23.0%)	<0.0001
	C	22 (62.9%)	11 (18.0%)	<0.0001
*L. crispatus*	V	15 (42.8%)	16 (26.2)	0.074
	C	13 (37.1%)	12 (19.7%)	0.052
*L. gasseri*	V	17 (48.6%)	17 (27.9%)	0.035
	C	17 (48.6%)	12 (19.7)	0.0033
*L. iners*	V	21 (60.0%)	12 (19.7%)	<0.0001
	C	19 (54.3%)	11 (18.3%)	0.0003
*L. jensenii*	V	7 (20.0%)	3 (4.9%)	0.026
	C	7 (20.0%)	2 (3.3%)	0.010
*L. vaginalis*	V	7 (20.0%)	4 (6.6%)	0.051
	C	6 (17.1%)	2 (3.3%)	0.026
*S. sanguinegens*	V	1 (2.9%)	2 (3.3%)	0.70
	C	1 (2.9%)	2 (3.3%)	0.70
*S. agalactiae*	V	8 (22.9%)	11 (18.0%)	0.38
	C	7 (20.0%)	5 (8.2%)	0.088
*M. curtisii*	V	5 (14.3%)	29 (47.5%)	0.0008
	C	4 (11.4%)	13 (21.3%)	0.17
*F. nucleatum*	V	12 (34.3%)	25 (41.0%)	0.33
	C	6 (17.1%)	17 (27.9%)	0.18

## Data Availability

Not applicable.

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
