# Peer review of "Vaginal and Cervical Microbiota Composition in Patients with Endometrial Cancer"

_ijms, 2023, doi:10.3390/ijms24098266_

Round 1

Reviewer 1 Report

In this manuscript, the authors sought to identify the changes in vaginal and cervical microbiota in relation to endometrial cancers. The authors have used PCR analysis to measure the levels of microorganisms in these tissues and identified statistical differences in overall and strain-specific microbiome  within these tissues in benign and endometrial cancer samples. Although this is interesting, their analysis is highly correlative and the study warrants more molecular analysis to conclude any relationship between the microbiome levels and endometrial cancers. In addition, can the authors find an association of the microorganism levels and patient survival? These lines of investigations will add better scientific foundation for their study.

Author Response

Dear Reviewer,

Thank You for Your comments which will improve quality of our paper. Methodology of our research included real-time PCR assays using 16S rRNA gene as the target. It is commonly accepted and widely used method of microbiome investigation presented in many scientific papers. We agree that the results are quite correlative and could be further improved with the use of more expanded methods, especially epigenetic analyses. Indeed, as a next step of our research, we plan to perform more detailed genetic investigations in cervicovaginal microbiome of endometrial cancer patients in close future. Thus, our conclusions are only speculative and not definite.

In regard to Your second suggestion: we have performed initial survival analysis, which proved no statistical correlation. It is mainly due to the fact, that endometrial cancer patients have quite good 5-year survival rates and vast majority of them are still alive. Most probably, to confirm any survival correlations, correlation reassessment should be performed no sooner than another 5-year period from now.

Reviewer 2 Report

With respect to the manuscript “Vaginal and cervical microbiota composition in patients with endometrial cancer” I think that the subject is interesting and that it can give an important contribution in health area.

Nevertheless, I would like to point out some aspects:

-It would be required minor spell check (for example, line 89 and line 312)

-The results section could be sudivided by topics to a better understanding

-The influence of epigenetic mechanisms could be explored

Author Response

Dear reviewer 2, thank You for Your comments improving quality of our paper. Your suggestions were taken into consideration and changes were provided:

-It would be required minor spell check (for example, line 89 and line 312)

Some errors were corrected including line 89, 141, 175, 312

-The results section could be subdivided by topics to a better understanding

The Results section was further subdivided into 7 subsections

-The influence of epigenetic mechanisms could be explored

We shortly discuss the problem of microbiome-induced epigenetic mechanisms in Discussion section. However, literature data is very limited.

Reviewer 3 Report

Congratulations

I really enjoyed reading your paper. It was well-designed and could expand our knowledge on cancer formation. I cannot found something in particular to be improved.

Great work!

Author Response

Dear Reviewer,

Thank You for Your support.

Round 2

Reviewer 1 Report

None